# Comparison of Achievable Contrast Features in Computed Tomography Observing the Growth of a 4H-SiC Bulk Crystal

**DOI:** 10.3390/ma12223652

**Published:** 2019-11-06

**Authors:** Michael Salamon, Matthias Arzig, Peter J. Wellmann, Norman Uhlmann

**Affiliations:** 1Fraunhofer Development Center X-ray Technology EZRT, 90768 Fürth, Germany; 2Crystal Growth Lab, University Erlangen-Nuremberg, 91058 Erlangen, GermanyPeter.Wellmann@fau.de (P.J.W.); Norman.Uhlmann@iis.fraunhofer.de (N.U.)

**Keywords:** computed tomography, PVT growth process surveillance, 3D in situ analysis, helical-CT, Bragg diffraction

## Abstract

Today the physical vapor transport process is regularly applied for the growth of bulk SiC crystals. Due to the required high temperature of up to 2400 °C, and low gas pressure of several Mbar inside the crucible, the systems are encapsulated by several layers for heating, cooling and isolation inhibiting the operator from observing the growth. Also, the crucible itself is fully encapsulated to avoid impurities from being inserted into the crystal or disturbing the temperature field distribution. Thus, once the crucible has been set up with SiC powder and the seed crystal, the visible access to the progress of growth is limited. In the past, X-ray radiography has allowed this limitation to be overcome by placing the crucible in between an X-ray source and a radiographic film. Recently these two-dimensional attenuation signals have been extended to three-dimensional density distribution by the technique of computed tomography (CT). Beside the classic X-ray attenuation signal dominated by photoelectric effect, Compton effect and Rayleigh scattering, X-ray diffraction resulting in the crystalline structure of the 4H-SiC superimposes the reconstructed result. In this contribution, the achievable material contrast related to the level of X-ray energy and the absorption effects is analyzed using different CT systems with energies from 125 kV to 9 MeV. Furthermore the X-ray diffraction influence is shown by the comparison between the advanced helical-CT method and the classical 3D-CT.

## 1. Introduction

Latest results of the in situ computed tomography (CT) of bulk 4H-SiC crystals [1] show that it is possible to analyze the three-dimensional surface geometry of 3” boules as shown in Figure 1 even during growth applying the Physical Vapor Transport (PVT) process. 

Due to the high penetration depth of the 3” SiC boule in combination with the relatively low X-ray energy applied, the achievable image quality in terms of density fidelity and contrast resolution is questioned. It is expected that the detection of voids or surface inhomogeneity is limited at the high penetration regions of the crystal. In previous contributions [2,3], the image quality of the in situ CT scan has been addressed, considering the peripheral furnace and vacuum chamber structure. The scope of this study explains if the image quality achievable for the bulk crystal provides a quality measure for the achievable results and provides an outlook for the further development of the CT technique to possibly increase X-ray energies and crystal sizes. 

## 2. Materials and Methods 

The achievable contrast in an X-ray image can be limited by an insufficient penetration, or a too small material thickness difference of the scanned object. Insufficient penetration is a main issue, inhibiting even high material thickness differences from detection behind a 3“ SiC boule. In order to analyze this influence the absorption behavior can be theoretically derived based on the attenuation coefficient and compared to an experimental measurement in a 2D radioscopy of the boule. The effect on the resulting image quality and the effective attenuation properties can be observed by an ex situ CT on a sample setup with corresponding X-ray energies. By applying the different X-ray energies shown in Table 1 to a special test setup the contrast and density detectability of the PVT-CT system is investigated. Except for the high energy scans, the voxel-size and scan time of the demonstrated CT method are related to the typical growth rate of approximately 100 µm/h. They result in a voxel-size of 130 µm and 30 min scan time, leading to a subpixel growth per CT scan. 

### 2.1. Attenuation Properties and Influence on the Density Fidelity in CT

The term "resolution" in CT can be distinguished in two aspects: The spatial and contrast resolution, which concurrently occur. The aspect of density fidelity extends the quality criterions of CT to the quantity of how homogeneous a certain material is resolved in the volume. The X-ray attenuation properties of a material are responsible for this effect and can be described by the Lambert–Beer law. The material type and its thickness, the attenuation coefficient µ and the unattenuated intensity I0 need to be considered to calculate the theoretical X-ray transmission. The attenuation coefficient is corresponding to the different X-ray energies and has been evaluated using XCOM, photon cross section database [4]. An applicable way to measure the attenuation behavior practically is to use a step wedge of the corresponding material with well-defined steps and X-ray energy. Instead of manufacturing a step wedge, the comparison of the theoretical with practical results has been done using a bulk SiC crystal with a convex shape. Its convex shape shown in Figure 2 leads to a variation of material thickness along the vertical plane in the obtained X-ray image. It allows measuring the transmission at different material thicknesses, similar to a wedge without steps with a known thickness. Due to the natural shape of the crystal, the determination of material thickness was performed using a reconstructed 3D volume. Each slice of the volume represents a cross-section of the crystal. Using a bounding box the elliptical crystal was measured in X and Y direction. Thus, with every cross section sampled, a mean material thickness was determined every 100 µm. This thickness variation was used as a virtual step-wedge for the calculation of the theoretical data as for the corresponding intensity value measured in each pixel of the transmission image. The 2D image data was obtained in one orientation and always the same orientation for the different energies to allow for the intensity values to accurately be related to the step thickness. Nevertheless, due to further effects like beam hardening, scattering and misalignments a slight discrepancy between the theoretical and practical values can be expected. Due to the focus of the investigation on the maximum achievable penetration and the resulting intensity distribution at high material thicknesses these influences are neglectable. 

### 2.2. Contrast Properties in Ex Situ CT Scan

The ex situ CT investigation is based on a special sample setup consisting of a graphite crucible with an inner diameter of 100 mm and an inserted 3” SiC boule with a convex shape as has been described already. In Figure 2, the different thicknesses analyzed are highlighted in the cross-sectional scheme by the dotted green lines. For contrast indication, eight threaded holes inside the crucible wall with a diameter of 6.7 mm oriented alongside the inserted SiC crystal were used. Depending on the total penetration depth at the different layers 1–3 of the crystal and the X-ray energy, the reconstructed shape of the round holes is affected as shown in Figure 3. The roundness of each hole inside the sample is determined by the different absorption behavior in the outer and inner beam path. The round shape is a suitable measure for penetration capability of the radiation.

This effect also indicates that a detection of structures in the affected layers is insufficient, and thus also inhibits the detection of voids and geometry features of the crystal itself. Therefore, a comparison of the crystals shape at different energy levels has also been obtained by applying a 3D variance analysis to the reconstructed volume data sets. The evaluation was performed on the 125 kV and 220 kV due to a most similar volume resolution and object orientation. The scans were performed on one and the same X-ray system without repositioning of the object in between. This approach minimizes efforts for CT geometry registration and avoids systematic errors deriving from resolution variances. The calculation was performed using the software VG-Studio Max 2 (Volume Graphics GmbH, Heidelberg, Germany). 

In order to analyze the contrast behavior different X-ray spectras were applied. Since the available X-ray energy corresponds to the design of a CT system, three different industrial CT types shown in Table 1 were used at the Fraunhofer Development Center for X-ray Technology (EZRT). The minimum energy applied corresponds to the reference PVT-CT setup described in [1]. The main difference between the CT systems is the type of X-ray source and its penetration and resolution capability. Whereas the micro CT uses a microfocus X-ray tube allowing high resolution down to several microns for small object sizes, the macro CT includes a minifocus X-ray tube. The sub-millimeter sized focal spot of this tube inhibits a magnified projection of the object and reduces the achievable resolution to the size of the detector’s pixel of approximately 150 microns. In the Linac CT the focal spot is even larger. For comparison reasons, all scans, except for the Linac CT, have been performed at a magnification factor of approximately 2, corresponding to the reference PVT-CT setup. The differences in sharpness resulting from the different focal spot sizes have not been considered. All scans were performed using 1200 projections on an angular scan range of 360°. Compared to the reference PVT-CT, this is approximately three times more than the standard in situ scan setting, but it allows for a better signal quality for the comparison. The processing of all projection data has been performed using Fraunhofer reconstruction library based on Feldakamp Davis Kress (FDK) [5] cone beam reconstruction algorithm. 

### 2.3. Side Effects of X-ray Diffraction

In addition to the main attenuation characteristics, X-ray diffraction at crystal lattice is unaviodable due to the crystaline state of the 4H-SiC. According to the Bragg diffraction principle [6], the incident X-rays are diffracted in an X-ray energy dependent angle. Some distinct energies superpose each other constructively at the lattice of the crystal. This superposition leads to so-called Bragg peaks being acumulated to two-dimensional diffraction patterns as indicated by the arrows in the 2D projection image in Figure 4. In crystalography the Bragg diffraction allows the crystal type (via measuring the lattice spacing) and lattice orientation to be determined. However, the boundary conditions regarding X-ray energy and sample illumination differ strongly from those applied in CT. Thus the patterns measured in CT cannot be quantitavely used in this setup. The Bragg diffracted intensity affects the resulting image quality due to the irregular occurance of the patterns in the attenuation signal. The investigation of the effect is focussed on the origin of the effect and it’s influence on the density fidelity in the standard PVT-CT setup. By observing the distribution of the patterns in 2D transmission images at different X-ray energies, the acceptance energy levels can be derived. Due to the angular sensitivity of the Bragg diffraction the images were obtained from 360° object rotation in 10° steps. The analysis was done by observing the intensity of the Bragg patterns in total based on a summation of maximum values of all projections to one image. Due to the symmetrical shape of the crystal, the occurance of the patterns can be observed well in the surrounding area and within the attenuation region. A bandpass filtering of the resulting data allows for a simultaneous visualisation of the surrounding and attenuation regions. Furthermore the influence of these effects on helical-CT scan and standard cone-beam CT (3D-CT) is analyzed in terms of artifact occurance. 

## 3. Results

### 3.1. Attenuation Properties and Influence on the Density Fidelity in CT

The comparison of the theoretically and experimentally derived attenuation properties is shown in Figure 5. The diagram shows the ratio of the transmitted X-rays at a corresponding material thickness. The theoretical attenuation values are related to a single X-ray energy, whereas the experimentally acquired results are based on a polychromatic energy distribution. Thus, the experimental signal distribution corresponds to an average X-ray energy level defined by the maximum acceleration of the electrons inside the X-ray tube given by the tube voltage, the Tungsten target and the pre-filtration setting described by the thickness of an output window and metal filter sheet. In general, all experimentally acquired results correspond to a much lower X-ray energy of the theoretical data. The 125 kV and 1 mm Tin (Sn) setting corresponds best to a 60 kV mono-energetic attenuation characteristic, the 220 kV and 2 mm Sn setting to the 100 kV mono-energetic and the 450 kV and 1 mm Sn to a 200 kV. The best correlation between the different graphs is in the region between 1 and 4 cm SiC thickness. Here, the experimentally achieved data corresponds to a regular e-function as described by the Lambert–Beer law. In the surrounding regions from 0 to 1 cm and from 4 to 8 cm, the correlation is disturbed. The main reasons for these deviations are unconsidered physical effects in the applied theoretical calculation. At first, the average energy of the X-ray spectrum changes along the penetration depth of the object: With higher thickness, a higher number of low energetic than high energetic photons are absorbed leading to an increased average energy, also called beam hardening. Secondly, a displacement of the theoretically assumed material thickness and the experimentally derived thickness from the CT scan might lead to a displacement of the graphs. Finally, at the high thickness range above 4 cm, the most dominant effect occurs: It seems that the experimental transmission ratio is much higher than the theoretical value, for example at 7 cm the transmission of the 125 kV 1 mm Sn spectrum is assumed to be approximately 58% higher than calculated with the 60 kV theoretical value. The same effect can be observed for all spectra. The deviation in this section of the graphs misleads to a much lower resulting absorption signal than expected by the Lambert–Beer law. The effect can be led back to the scattered radiation occurring especially when using a cone beam X-ray setup. While the theoretical attenuation values are related to a scattering free, single photon energy and single pixel measurement, scattering is not considered. In the experimental setup, a cone-beam geometry illuminating the whole bulk crystal was applied, leading to a widespread scattered radiation all over the detector. Whereas in regions of high transmission the intensity of scattered radiation is just a relatively small fraction of the total signal, it becomes dominant in the regions of low transmission signal. This is the case for all applied X-ray energies. 

This behavior affects the density fidelity, especially along the different crystal layers as shown in Figure 6 by a line profile extracted from a CT cross-section of the crystal at different energies. In the low X-ray energy regime (125kV), which is used in the PVT reactor setup, the high attenuation coefficient variation from facet to the seed displayed by the blue dots also misleads to a fluctuating density. At the 550 kV scan the fluctuation is reduced due to better penetration capability and less beam hardening artefacts. Nevertheless, even here, the density value still differs inhibiting the analysis of i.e., density distribution inside the crystal volume.

### 3.2. Contrast Properties in Ex Situ CT Scan

The results of the CT scans performed using the same sample on different CT systems and energies are shown in Figure 7. For each X-ray energy regime, three representative cross-sections with a similar crystal diameter were extracted. One of the threaded holes has been digitally zoomed to allow for a better comparison. The reconstructed shapes of the threaded holes at different crystal layers shown in Figure 2 differ depending on the penetration length and the applied X-ray energy. While at high energies, represented by the two lower image rows, the shape remains nearly circular in all thickness layers, it forms out elliptical with larger thickness and lower energy shown in the two upper image rows. This effect is caused by the discrepancy of the transmission signal between the tangential and the central beam path shown in Figure 3. As mentioned before, in the attenuation characteristics of SiC, the total absence of the transmission signal in the longest penetration areas leads to a directionally smeared edge of the holes. The smearing distributes perpendicular to the central beam path because of the missing information in the center path. The effect applies not only at the surrounding peripheral structure but also at the crystal itself.

The quantification of this influence on the crystal itself is shown in Figure 8. By the variance analysis between the 125 kV and 220 kV datasets, the effect can be confirmed. The 3D distribution of the variance shown contains just minor deviations (marked green) in the short penetration region under 50 mm and higher deviations (marked magenta and violet) of up to +/- 0.2 mm in the regions at maximum crystal diameter. 

### 3.3. Side Effects of X-ray Diffraction

Figure 9 shows different Bragg patterns at three different X-ray energy settings. The intensity appearance of the signal patterns at the different energies allow to quantify the energy range which is superimposed by the crystal lattice. The images shown at 60 kV (a), and at 125 kV (b), were obtained using an additional beam collimation in order to better recognize the patterns in the surrounding region. Without collimation, the superposition of the unattenuated intensity leads to a low contrast between the peaks and the X-ray intensity. The still visible patterns at 60 kV indicate that the energy, leading to a constructive interference, lies in the low energy section. Typically, the Bragg peaks are produced by the high intensity characteristic lines of the X-ray spectrum, leading to the assumption that the L-lines of the Tungsten target of the X-ray tube, with distinct energies between 8 and 11 keV, are responsible for most of the patterns in the surrounding region. This is also confirmed by the scan performed at 220 kV and a relatively strong pre-filtration of 2 mm Tin shown in Figure 9c. At least the patterns in the surrounding regions are strongly reduced. In the center region of the crystal some patterns still exist. The central location implicates a higher acceptance energy level with a correspondingly smaller Bragg diffraction angle. The influence of the patterns on CT reconstruction depends on the type of trajectory chosen. In Figure 10, two cross sections of the ex situ helical-CT scan (a), and a 3D-CT scan (b), are compared. The scan parameters correspond to the standard values with 125 kV and no additional pre-filtration. The helical result in Figure 10a shows needle type structures in the bulk SiC crystal. Based on the attenuation and contrast properties of the bulk crystal previously described in 3.1, we come to the conclusion that none of the structures are real. Due to the trajectory type of helical-CT, the patterns are spread at varying imaging angles leading to a different behavior than in standard 3D-CT where the trajectory remains at one plane of the crystal during the whole scan. For the analysis of the influence of the Bragg patterns on the in situ CT scan, the SiC boule has been inserted into the graphite crucible. 

In Figure 11, the result of the in situ crucible setup is shown. Even though the image was acquired with the same parameters as the result shown in Figure 9b, the intense patterns are mostly suppressed. This reconfirms the conclusion of quite low X-ray energies accepted by the lattice of the SiC. The energies are low enough to be fully absorbed by the graphite crucible surrounding the crystal. 

## 4. Final Discussion

The results show that even for the ex situ case—not considering the furnace periphery of the in situ PVT reactor—the attenuation characteristics related to the investigated sample and the applied X-ray energy are quite challenging. The main demands derive from the size of the SiC boule leading to a reasonably high energetic X-ray source. The latter also results in high efforts for radiation protection and duty costs. Whereas for lower crystal diameters under 50 mm relatively low energies of 125 kV can be applied, for the larger diameter, even the double value does not suffice. The contrast analysis at different energies leads to the conclusion that under 220 kV, the detection of bulk SiC with diameters above 50 mm is solely related to the tangential beam paths, and thus do not contain information about structures potentially appearing in the center of the crystal such as voids. Respectively, density variations are not measurable in these regions. Nevertheless, it is unexpected that the surface shape can be reconstructed very well in most regions, even with the lowest energy applied. The deviations measured by the variance analysis differ regionally but do not exceed the range of 2–3 voxels and are thus well usable for the surveillance of growth, even in the short term. The reason for this behavior can be found in the investigation of attenuation characteristics. Due to the predominance of scattering at the high thickness crystal regions shown in Figure 7, the reconstruction is not as negatively affected as it would be by a zero value occurring at the longest penetration depth in an ideal theoretical case. The pre-reconstruction step of logarithmic calculus needed to linearize the attenuation signal in the 2D image would lead to a higher attenuation coefficient the lower the signal behind the crystal is. The result would be an oversaturated region in thickness layers with pure intensity. The scattered intensity avoids that. The second aspect considers the structural resolution of the crystal shape. The symmetrical shape of the crystal without large undercuts inhibits artifacts occurring when the 360° information of the object is inconsistent, as shown by the round shaped holes. The continuous and smooth surface of the SiC boule is conductive. The resulting density fidelity of the CT method is not only affected by the missing transmission signal. It also suffers from further non-linearities like beam hardening and scattering, resulting in a signal gradient within the homogeneous SiC boule. As shown in Figure 6, the intensity is increased even in the lower penetration regions leading to a halo type effect close to the surface of the crystal. The results of the Bragg diffraction and especially those related to the in situ measurements are very promising, as due to the peripheral structure of the crucible, furnace and insulation layers, most of the patterns are reduced. Especially the low energetic but high intense Bragg patterns seem to be absorbed well before being imaged in the in situ approach. The high energetic fractions appearing within the attenuation area of the crystal, due to their small Bragg angle, might still have some relevance and will be analyzed further in future work.

## 5. Conclusions

The conclusion of the work is that the application of low energy X-radiation already provides a very good imaging capability at low complexity of the X-ray system described in [1]. Even an enhancement of the X-ray energy by the factor of two could not satisfy the required penetration capability for a total transmission. The application of even higher energies is less practical due to the size of the components and the high shielding efforts. Additionally, frequent scanning during the continuous growth process allows capturing the layers at lower thicknesses during growth. Due to the convex growth front, each layer of the crystal can be scanned once with an appropriate image quality before it is overgrown. Effects like beam hardening, scattering and Bragg diffraction affect the density fidelity. With these conclusions even an extension of the method to 6” and 8” SiC boules is supposed to be practical, opening a wide application field in today’s rapidly emerging field of crystal growth by PVT.

## Figures and Tables

**Figure 1 materials-12-03652-f001:**
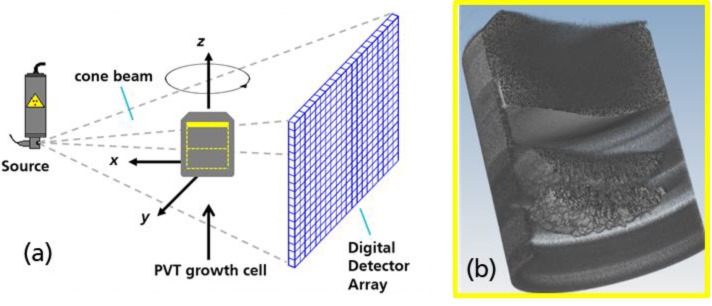
**PVT-CT setup,** (**a**) Computed tomography (CT) principle applied to the crystal growth. (**b**) Resulting CT volume measured in situ, reconstructed and 3D rendered with an opened cross-section plane.

**Figure 2 materials-12-03652-f002:**
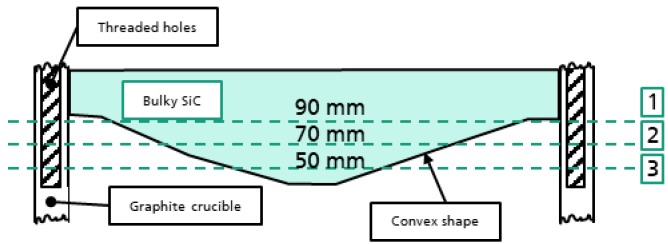
Scheme of the sample setup consisting of the convex shaped bulk SiC crystal and the surrounding graphite crucible. The different thickness layers relate to the convex shape of the crystal. The dotted lines indicate certain thicknesses analyzed in the reconstructed volume.

**Figure 3 materials-12-03652-f003:**
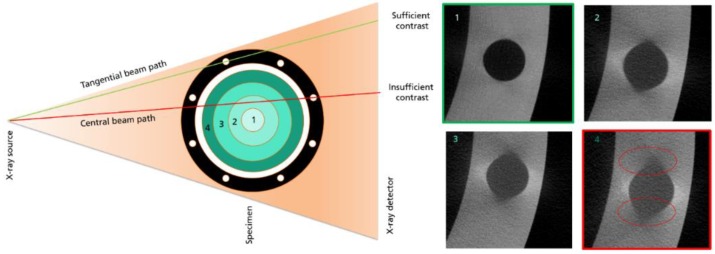
The schematic principle of the contrast properties in different beam positions and at different thickness layers (1–4). With a larger crystal diameter, the detection of the round shaped hole is affected.

**Figure 4 materials-12-03652-f004:**
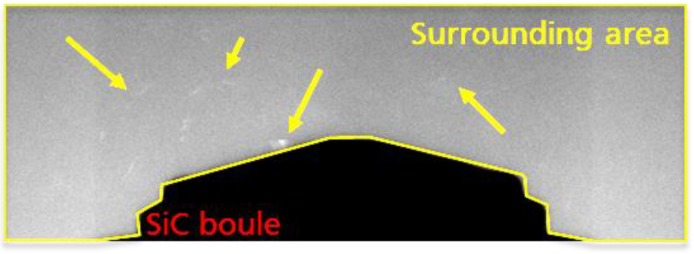
Definition of the different image areas affected by the Bragg diffraction patterns. The arrows in the surrounding area indicate the single peaks characterized by a higher intensity. Behind the SiC boule attenuation area (dark) the patters are also visible.

**Figure 5 materials-12-03652-f005:**
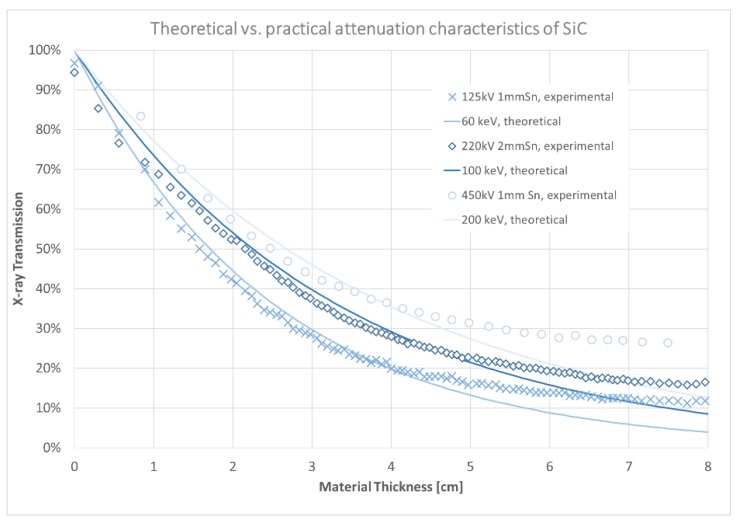
Comparison of the theoretical and experimental attenuation characteristics of SiC at different X-ray energies.

**Figure 6 materials-12-03652-f006:**
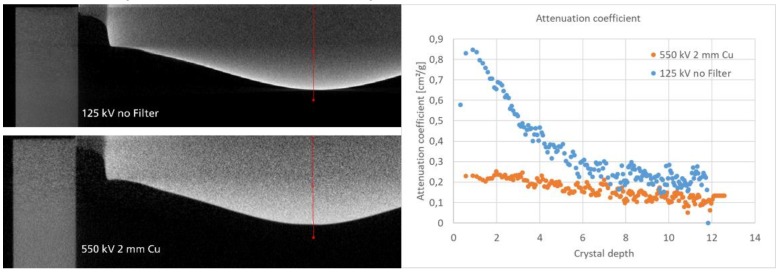
The influence of attenuation properties and X-ray energy on the density homogeneity in the reconstructed cross-section. The plot shows the two line profiles from bottom to top.

**Figure 7 materials-12-03652-f007:**
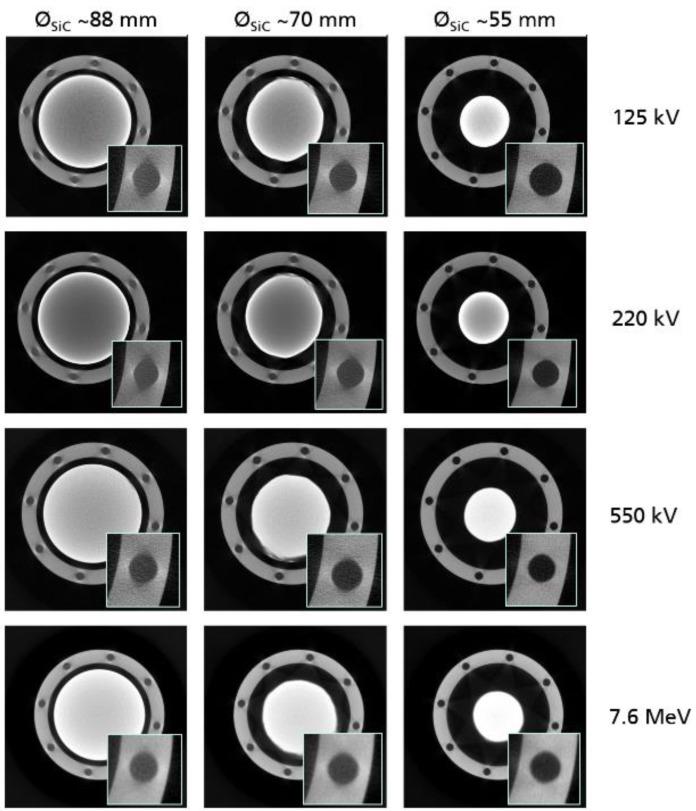
Comparison of the different CT scan results obtained using different X-ray energies. Each row represents a cross-section with a varying crystal diameter leading to different quality of reconstruction. The digitally zoomed hole indicates the contrast capability of the different settings.

**Figure 8 materials-12-03652-f008:**
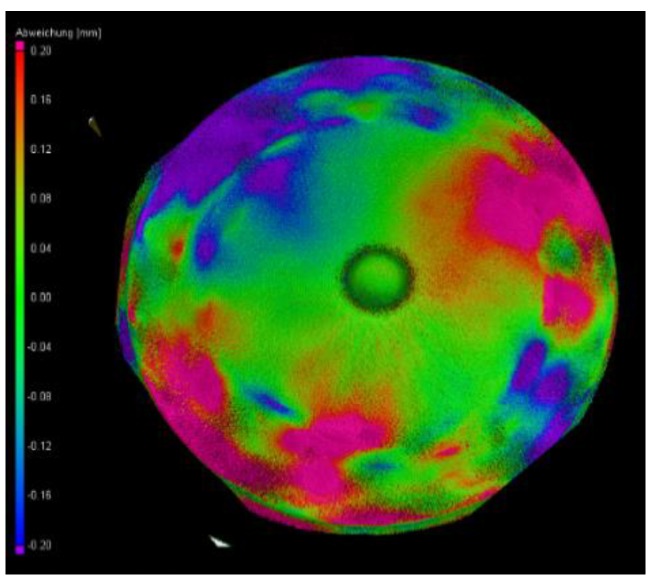
Variance analysis of the 125 kV and 220 kV scans. The colors code the difference in the range of approx. +/- 0.2 mm.

**Figure 9 materials-12-03652-f009:**
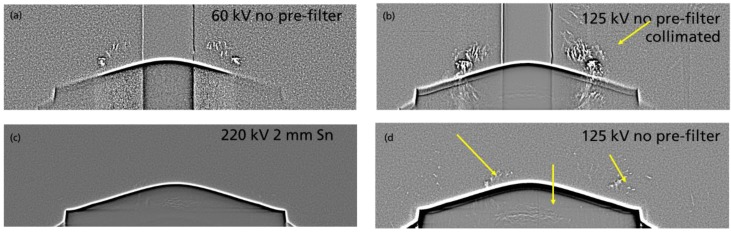
Brag diffraction patterns superposing the 2D projection images acquired by averaging images from 360°. The different images were acquired at different X-ray energies. The images have been filtered using a bandpass filter. Images (**a**,**b**) were obtained using a vertical lead slit for collimation. This approach allows a better detection capability of the peaks. Without collimation some of the signals are superposed by the unattenuated X-ray intensity as can be seen comparing (**b**,**d**). The cut-off of the low energies applied in (**c**) confirms that diffraction is induced at low energies.

**Figure 10 materials-12-03652-f010:**
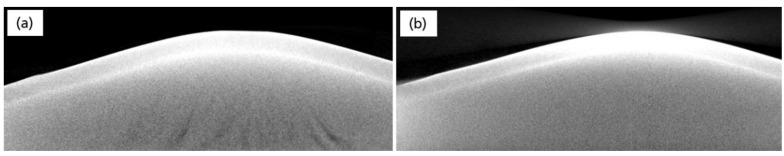
Comparison of the Bragg influence on different CT modalities in the ex situ case. Image (**a**) shows a helical-CT scan with additional contrast appearing inside the bulk crystal. Image (**b**) shows a standard 3D-CT scan without the effects. Both scans were acquired at identical X-ray energy settings at 125 kV and no additional pre-filtration.

**Figure 11 materials-12-03652-f011:**
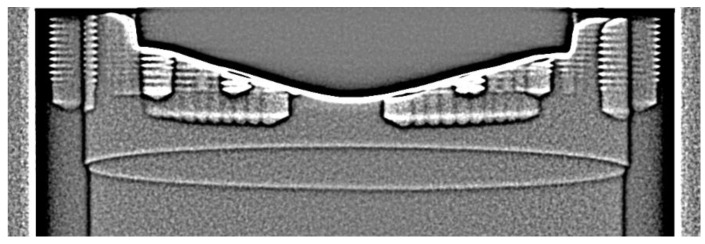
In situ setup of the crucible scanned at 125 kV and no pre-filter. None of the patterns affect the image due to the filtering effect of the crucible walls.

**Table 1 materials-12-03652-t001:** The applied computed tomography systems and parameters of the different investigations.

Name	Max X-ray Energy	Applied X-ray Energy	Pre-Filtration (mm)
PVT-CT	125 kV	125 kV	PVT Reactor ^1^
Micro CT	225 kV	125 kV	1 Al
Micro CT	225 kV	125 kV	1 Sn
Micro CT	225 kV	220 kV	2 Sn
Micro CT	225 kV	220 kV	0.5 Cu
Macro CT	600 kV	450 kV	1 Sn
Macro CT	600 kV	550 kV	2 Cu
Linac CT	9 MV	7.6 MV	2 Al

^1^ PVT reactor represents the standard values applied during growth

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
