# Peer review of "Comparison of Achievable Contrast Features in Computed Tomography Observing the Growth of a 4H-SiC Bulk Crystal"

_materials, 2019, doi:10.3390/ma12223652_

Round 1

Reviewer 1 Report

Good work and should be valuable to CT scan users in general as well as those with a specific interest in SiC crystal growth. A couple of minor things requiring no more than editorial changes. (1) Add comments on scan time scale versus crystal growth time scale. Is the crystal expected to change in size and/or shape appreciably (compared to spatial resolution or pixel size) while being scanned (CT can take anything between mins to hours depending on settings). (2) Figure 5. Not clear which are theoretical and which are experimental. Perhaps change theoretical to line and leave experimental as symbols? Why transmission at zero-thickness is 100% for some and <100% for others? (3) Figure 6. Can SiC crystal be considered a homogeneous solid mass? If so, as long as one can distinguish between the solid and the background, SiC crystal growth can be monitored. If SiC cannot be considered homogeneous, adding a filter to reduce (but not completely remove) beam hardening effect doesn’t really help that much, since the filtered result isn’t uniform either. (4) There is a typo in Line 130: “beeing” should be “being” I think.  

Author Response

Thank you for the comments, here are my answers:

(1) I have added the following text in line 60:

Except the high energy scans, the voxel-size and scan time of the demonstrated CT method are related to the typical growth rate of approx.. 100 µm/h. They result in 130 µm and 30 min. scan time leading to a subpixel growth per CT.

(2) I have reworked the diagram. Thanks also for the comment on the irregularities concerning the zero values. After checking the diagram I've found a mismatch of the scale for some theoretical values. The remaining inconsistency at the experimental values are related to the edge soread function in the transition region between crystal and air.

(3) You're correct, the main aspec is fullfilled with 125kV assuming the crystal to be solid. By the comparison we we wanted to show that density changes are not easily measureable even at very high X-ray energies and with extensive system costs... I have added the following paragraph:  Nevertheless, even here the density value still differs inhibiting the analysis of i.e. density distribution inside the crystal volume.

(4) found and fixed.

Thank you!

Reviewer 2 Report

This manuscript provides very important and interesting results of in-situ monitoring using X-ray of the physical vapor transport (PVT) growth process of SiC single crystals. The authors deliberately examined the X-ray beam conditions and demonstrated that the proposed method makes it possible to detect the growth front shape of SiC crystals during PVT growth. The obtained results and infomartion would lead to fundamental understanding of the PVT growth process and will be of great interest to the SiC commnity. Overall, the manuscript would merit publication, provided that revisions as to the following points are made.

(1) In Fig. 10, the energy of the X-ray beam used for this experiment should be provided in the figure or the caption.

(2) Line 257: “Figure 10” could be “Figure 11”.

Author Response

Thank you for the comments.

(1) I have added the energy value in the caption

(2) found and fixed

Thanks

Reviewer 3 Report

In this work, we propose an original method for controlling the
growth process of a bulk SiC crystal - X-ray tomography.
It seems to me that this method is promising for practical
application and can give a lot of new information about the
process of crystal growth. Earlier, I had not heard anyone use
such a measuring technique. Therefore, I believe that this work
is new, relevant and important for practical application.
I believe that it can be published in its current form.

Author Response

Thank you for the review.